# Bloodletting for Acute Stroke Recovery: A Systematic Review and Meta-Analysis

**DOI:** 10.3390/healthcare12202060

**Published:** 2024-10-17

**Authors:** Mikyung Kim, Changho Han

**Affiliations:** 1Department of Internal Medicine, Dongguk University Ilsan Oriental Hospital, Goyang-si 10326, Republic of Korea; 2Department of Internal Medicine, Dongguk University WISE Campus, Gyeongju-si 38066, Republic of Korea; changho.han@gmail.com

**Keywords:** bloodletting, blood pricking, cerebrovascular accident, cerebrovascular disorders, stroke, traditional Chinese medicine, traditional Korean medicine, meta-analysis, systematic review

## Abstract

Background: Bloodletting is a non-pharmacological treatment commonly used for acute stroke in traditional East Asian medicine. This study evaluated the efficacy and safety of bloodletting in acute stroke recovery. Methods: We conducted a comprehensive search of eight electronic databases up to 4 June 2024 to identify relevant randomized controlled trials (RCTs). Review Manager 5.4 was used for the meta-analysis, with methodological quality assessed using the Cochrane Risk of Bias 2 tool and the GRADE approach. Results: Seventeen RCTs were included in this meta-analysis. The bloodletting group showed statistically significant improvements in neurological deficits compared to the non-bloodletting group, as measured using the National Institutes of Health Stroke Scale (mean difference [MD]: −2.08, 95% confidence interval [CI]: −3.13 to −1.02) and the treatment effective rate (risk ratio: 1.17, 95% CI: 1.11 to 1.22). Motor function also improved significantly in both upper (Fugl-Meyer Assessment, MD: 12.20, 95% CI: 9.67 to 14.73) and lower extremities (MD: 3.86, 95% CI: 2.16 to 5.56). The effect on daily living activities was not significant overall, but benefits were observed in patients treated within three days of stroke onset (Barthel Index, standardized MD: 0.85, 95% CI: 0.01 to 1.69). No significant differences in the frequency of adverse events were observed between the groups. Conclusion: Bloodletting may be an effective and safe adjunctive therapy for patients with acute stroke receiving conventional Western medical treatment. However, further research is necessary because of the small sample sizes and low quality of the included studies.

## 1. Introduction

Stroke is a non-communicable disease managed intensively by the World Health Organization because it has long-term effects, is difficult to cure, and is a major cause of death worldwide [1]. Although the mortality rate of patients with stroke is decreasing [1], many stroke survivors develop disabilities and complications lasting the rest of their lives [2]. It is a major cause of failure to live independently in daily life and return to work, thus lowering the quality of life of individuals and increasing their socioeconomic burden [2]. More than half of patients with acute stroke have multiple disabilities, which are known to be prognostic factors for long-term disability and death [3]. Despite remarkable advances in acute stroke treatment over the past two decades, only a fraction of patients who meet the stringent criteria for intervention are eligible for such treatment; however, stroke remains a leading cause of long-term disability worldwide [4]. This is the reason for the demand for traditional and complementary medicines in patients with stroke [5]. The utilization rate of traditional medicines in stroke patients is particularly high in East Asia [6,7,8].

Although it has been studied less than acupuncture, which is now world-renowned [9], bloodletting is also a major non-pharmacological treatment in traditional medicine widely used for acute stroke patients in cultures in this region, including Korea and China [10,11,12]. Bloodletting is a common treatment found in traditional medicine worldwide, including in East Asia [13]. The method of bloodletting varies according to region and era [13], but in East Asia including China and Korea, it is still performed by pricking specific acupoints with an acupuncture needle to release small drops of blood [12,14]. Many people in this cultural sphere still seem to firmly believe in the usefulness of bloodletting for acute stroke care [10,15]. Scientific explorations have been conducted on the possible efficacy and mechanism of bloodletting in acute stroke. Several preclinical studies have shown that bloodletting can improve various pathologies of acute brain injury, such as decreased blood–brain barrier permeability [16], increased nitric oxide synthase activity [16], and regulation of coagulopathy [17], thereby alleviating cerebral edema [16], improving cerebral blood flow, and exerting neuroprotective effects [18], which may ultimately result in clinical benefits such as improvement of neurological deficits.

Clinical studies suggesting the positive effects of bloodletting in patients with acute stroke have been published [12], and a systematic review (SR) and meta-analysis (MA) of related randomized controlled trials (RCTs) have been reported [19]. However, the previous SR included studies on the effects of bloodletting performed on only specific acupoints and reported localized and not universally used outcomes, such as total effective rate (TER) and Chinese Stroke Scale (CSS). They did not report the effect of bloodletting measured with widely used international stroke scales such as the National Institute of Health Stroke Scale (NIHSS) and Barthel Index (BI) or the quantitative analysis results for safety [19].

Therefore, we judged that it is necessary to re-assess whether bloodletting for acute stroke still has clinical benefits, even in the context of modern society wherein conventional treatment with Western medicine is commonly used. To achieve a more comprehensive review than the previous SR, we attempted an SR and MA that included more recently published RCTs without restrictions on the type of acupoint used, applying more universal outcomes in stroke assessment. Specifically, this study aimed to evaluate the efficacy and safety of bloodletting for acute stroke recovery.

## 2. Materials and Methods

### 2.1. Protocol and Registration

This SR and MA were conducted in compliance with the Preferred Reporting Items for SR and MA (PRISMA) 2020 statement (Appendix A) [20]. The study protocol was registered in the International Prospective Register of SR (PROSPERO: CRD42024571447).

### 2.2. Eligibility Criteria

In this section, we describe this study’s inclusion and exclusion criteria. (1) Participants: Studies targeting patients diagnosed with acute stroke based on brain imaging and clinical findings were included. All participants, regardless of group assignment, received standard care according to Western medicine for acute stroke, such as general supportive care, medications to prevent and manage complications, nursing interventions, and rehabilitation. Intravenous thrombolysis, endovascular therapies, and any surgical treatment were also allowed if eligible. There were no restrictions on age, sex, or ethnicity. (2) Intervention: Studies assessing the effects of bloodletting were included in this review. In this study, bloodletting was defined as pricking the skin at acupoints with sharp-tipped instruments, such as three-pointed acupuncture needles or skin needles, and releasing small drops of blood according to the context of East Asian traditional medicine [12,14]. Venesection, phlebotomy, and hemodilution were not considered in this review because, unlike these intervention techniques, which incise blood vessels and cause a substantial change in blood volume, the bloodletting defined in our study clearly differs in that it involves pricking the skin with a small needle to release only a few drops of blood. In addition, *Hijama* or wet cupping, which is performed by repeatedly pricking the skin with needles and then applying negative pressure to promote more blood discharge, was not considered in our study. This is because these interventions clearly differ from the bloodletting defined in our study in that they target a certain area rather than a single point and involve physical stimulation that induces negative pressure in addition to pricking. For the same reason, studies that combined bloodletting therapy with other interventions and could not detect the effect of bloodletting alone, such as herbal medicine plus bloodletting or acupuncture combined with bloodletting, were also excluded. (3) Comparator: No bloodletting, waiting list, or sham interventions were allowed. Western treatments for acute stroke as active control interventions were allowed. However, interventions that are not formally acknowledged as standard treatments for acute stroke in Western medicine, such as acupuncture, massage, herbal medicine, and other types of bloodletting, were not included as comparators. (4) Outcomes: The primary endpoint of our study was the presence of neurological deficits, measured using stroke assessment tools, that is, NIHSS, European Stroke Scale, or CSS. Studies that assessed the TER based on the degree of neurological deficit were also included. The secondary endpoints included measures of independence in activities of daily living (ADLs), motor function, stroke recurrence, and death. The frequency of adverse events (AEs) was also assessed to evaluate the safety of bloodletting. (5) Study design: Only RCTs were included. When data from the same participants were published in two or more papers, they were considered one study. When a clear discrepancy between the data presented in the paper and the sentences described in the text was identified, the study was excluded because of its unreliability.

### 2.3. Search Strategy

Papers published before the search date (4 June 2024) were included. Electronic databases and clinical research registration websites were searched from their inception as follows: Core databases (PubMed, Embase, and Cochrane Library), Chinese Academic Journals (CAJ), Korean databases (Science On, Korean Information Service System, KoreaMed, and Oriental Medicine Advanced Searching Integrated System), and clinical research registration platforms including International Clinical Trials Registry Platform (ICTRP), ClinicalTrials.gov, Chinese Clinical Trial Register (ChiCTR), and Clinical Research Information Service of South Korea (CRIS). Reference lists of the selected papers were manually searched. The following search terms were used, with the search strategies adjusted to suit each database: (stroke OR cerebral infarction OR cerebral hemorrhage OR cerebrovascular disorder OR cerebrovascular accident OR apoplexy OR brain ischemia) AND (bloodletting OR blood pricking OR collateral pricking OR fang xue) (Appendix A). No language restrictions were applied in this study.

### 2.4. Study Selection and Data Extraction

After the retrieved papers were imported into EndNote, duplicate papers were automatically and manually removed. Two reviewers reviewed the titles and abstracts and removed papers that were completely irrelevant to the topic of study. After obtaining and reading the full texts of the papers that passed this screening process, the reviewers selected only those papers that met the predetermined eligibility criteria. Data from the selected studies were extracted and entered into a Microsoft Excel spreadsheet (2019). The extracted data were as follows: basic information on the paper (first author’s name, publication year, and country), participant information (sample size, sex, age, stroke type, and time since stroke onset), details of the interventions, and outcome measures. The post-treatment measurement value was used as the outcome value. If there were multiple post-treatment measurements, the value measured first after the end of the treatment period was selected. All of the above processes were conducted independently by two reviewers, and any disagreements were resolved through discussion.

### 2.5. Risk-of-Bias Assessment

The Cochrane Risk of Bias (RoB) tool (version 2) was used to assess the internal validity of the included RCTs [21]. RoB 2 was designed to assess the following five domains: randomization process, deviations from the intended interventions, missing outcome data, measurement of the outcome, and selection of reported results. According to the results of each domain assessment, the overall RoB of a study was classified as “low”, “high”, or “some concerns”. Two reviewers independently assessed the RoB of each study, and any disagreements were resolved through discussion.

### 2.6. Data Analysis and Synthesis

We performed MA using Review Manager 5.4 software (RevMan) and derived forest plots. The effect size of the continuous data was calculated using the inverse variance method, and the mean difference (MD) or standardized MD (SMD) with 95% confidence intervals (CIs) was calculated. The effect size of the dichotomous data was calculated using the Mantel–Haenszel method, and risk ratios (RRs) with 95% CIs are presented. The random-effects model was used when Cochrane’s I^2^ value was more than 50%, reflecting statistically substantial heterogeneity between studies [22], and when the characteristics of the participants and interventions in the included studies were qualitatively heterogeneous. Otherwise, a fixed-effects model was considered.

Subgroup analysis was performed when three or more studies were included in the MA. Subgroups were determined according to stroke type, time since onset, treatment period, treatment interval, and total session, and the contribution of these factors to heterogeneity was explored.

To assess publication bias, the visual asymmetry of the funnel plots derived using RevMan (5.4) was evaluated when ten or more studies were included in the MA. In addition, Egger’s regression test was performed using R software (4.4.1) to reduce visual errors and explore small-study effects. The leave-one-out method was used for the sensitivity assessment.

### 2.7. Certainty-of-Evidence Assessment

The Grading of Recommendations, Assessment, Development, and Evaluations (GRADE) approach [23] was used to assess the quality of evidence derived from the MA in this study. Web-based GRADEpro was used to assess five domains: RoB, inconsistency, indirectness, imprecision, and publication bias. Based on the assessment results for each domain, the certainty of evidence was classified as high, moderate, low, or very low. Two reviewers independently performed the GRADE assessment, and any disagreements were resolved through discussion.

## 3. Results

### 3.1. Study Selection

A total of 477 records, including 459 from eight databases and 18 from four clinical trial registries, were searched. After excluding duplicate records, the remaining 425 articles were screened based on their titles and abstracts. Fifteen records that met the eligibility criteria were selected by reviewing the full texts of the remaining articles. Bibliographic information on the records excluded during this process is provided in Appendix A. Since two [24,25] of the fifteen selected records [24,25,26,27,28,29,30,31,32,33,34,35,36,37,38] were clearly based on data from the same participants, they were considered as one RCT, and only representative records [25] were cited in the text of this paper. In addition, three eligible RCTs [39,40,41] were identified through a manual search. Finally, 17 RCTs [25,26,27,28,29,30,31,32,33,34,35,36,37,38,39,40,41] from 18 records [24,25,26,27,28,29,30,31,32,33,34,35,36,37,38,39,40,41] were included in the MA (Figure 1).

### 3.2. Characteristics of the Included Studies

The characteristics of the 17 included studies [25,26,27,28,29,30,31,32,33,34,35,36,37,38,39,40,41] are listed in Table 1. These studies were published between 2005 and 2022. All of these studies were conducted in mainland China and published in Chinese in domestic journals. Most were two-armed studies except for five [27,32,36,38,39] with three arms. We extracted only the data of the paired groups that could confirm the effect of bloodletting within each study. As a result, we obtained data from two groups across all studies: the bloodletting and the non-bloodletting groups. All studies administered conventional Western medicine for acute stroke to all patients and added bloodletting only to the treatment group. None of the studies applied an active control group or a sham intervention. All included studies compared the bloodletting and non-bloodletting groups.

The number of participants in each study included in the MA ranged from 35 to 350. Most of these patients had ischemic stroke. One study [38] recruited only patients with hemorrhagic stroke, while another [31] recruited patients with either ischemic or hemorrhagic stroke. All of the studies included patients with acute stroke. Ten [25,26,27,31,32,34,36,37,40,41] enrolled patients within three days of onset. Treatment intervention began immediately after patient enrollment, except in one study [38] in which bloodletting was initiated after vital signs were stabilized after surgery.

The most commonly used acupoints for bloodletting were the *Jing*-Well acupoints. Except for two studies that selected ear apex acupoints [35] and Ex-UE 11 (*Shixuan* acupoints) [37], all others used more than one Jing-Well acupoint. All studies used acupuncture needles to puncture the acupoints. The amount of bleeding ranged from one to ten drops per acupoint, with the most common being three drops or less [26,27,28,30,31,32,34,36,38]. The treatment frequency ranged from three times per day [32] to once every three days [38]. Approximately once a day (including five to seven times a week) [25,26,27,30,31,33,35,36,37,39] was the most common. The total number of bloodletting sessions ranged from five [36] to eighty-four [32]. Ten [27,29,40], twelve [28,35,38], and fourteen sessions [25,31,39] were most common. The treatment period ranged from one week [26,36] to approximately seven weeks [38]. Two weeks was the most common [25,27,31,35,39,41].

Fourteen studies [25,28,29,30,31,32,33,34,35,36,38,39,40,41] provided TER data based on neurological deficits. Four studies presented the post-treatment measurement value [25,28,36] or the change between baseline and the end of treatment [38]. Eight studies [26,27,30,31,33,35,40,41] reported total post-treatment CSS scores. To assess independence in ADL, five studies assessed BI [25,27,28,33] or modified BI [37]. Four studies [30,36,38,40] used the Total Life Ability (TLA) score, but only two [30,36] provided post-treatment values. One study [25] presented this function using both BI and functional comprehensive assessment (FCA) scores. For motor function assessment, five studies [28,32,35,37,38] used subdomains of the Fugl-Meyer Assessment (FMA). Some of these studies also used muscle-power-related subdomains of the CSS [35,37]. No studies reported stroke recurrence or death.

For safety assessment, only three studies [25,28,41] provided information on AEs. One of them reported that no AEs were observed [41]. The other two [25,28] administered bloodletting combined with acupuncture to all subjects, and the most commonly reported AE was bruising in both the treatment and control groups. Among the reported AEs, the only serious one was pneumonia, which occurred in the control group. All of the other reported AEs were mild.

### 3.3. RoB Assessment

The results of the RoB assessment of the included studies are shown in Figure 2. In the first domain of the randomization process, the RoB of all studies was assessed as having some concerns. Except for nine studies [25,27,33,36,37,38,39,40,41] that used a random number table, the remaining studies did not provide any information on the randomization process. No studies reported information on allocation concealment. The RoB of all studies was assessed as low in the second domain for deviations from the intended interventions. This is because there were no cases where the preplanned intervention was changed during the trial, although no study explicitly described patient or practitioner blinding. The RoB of all studies was assessed as low in the third domain for missing outcome data. This is because data from almost all randomly allocated patients were analyzed in the Results section of each study. In the fourth domain of the outcome measurement, the RoB of all studies was high. This is because no study described the implementation of assessor blinding; if assessors were aware of the allocated groups, it could not be ruled out that this might have influenced the outcome assessment. In the last domain of the selection of reported results, the RoB of all studies was assessed as having some concerns. This is because no study provided evidence that could confirm the predetermined evaluation plan, such as a protocol paper or clinical trial registration. Thus, it was not possible to judge the consistency between the actual measurement and the reported results. Therefore, the overall RoB of all studies was high.

### 3.4. Meta Analysis

#### 3.4.1. Neurological Deficit

The pooled overall effect of the neurological deficits is presented in Figure 3. Bloodletting significantly improved NIHSS (three studies; MD: −2.08; 95% CI: −3.13 to −1.02) (Figure 3A), CSS (eight studies; MD: −4.15; 95% CI: −4.59 to −3.71) (Figure 3B), and TER (14 studies; RR: 1.17; 95% CI: 1.11 to 1.22) (Figure 3C). No statistical heterogeneity between studies was identified for this outcome (I^2^ = 0%, 0%, and 1%, respectively). However, subgroup analysis showed that the significant effect of bloodletting on NIHSS (MD: −2.38; 95% CI: −5.04 to 0.28) and CSS scores (MD: −3.60; 95% CI: −8.91 to 1.71) disappeared when the treatment period was one week or less (Appendix A). For NIHSS, the significant effect of bloodletting also disappeared even when the number of treatment sessions was 10 or fewer (MD: −2.38; 95% CI: −5.04 to 0.28) (Appendix A). The effect of bloodletting on improving TER was clearly significant only in studies including patients with cerebral infarction (Appendix A).

#### 3.4.2. ADL Function

The estimated effect from MA showed that bloodletting did not significantly improve the post-treatment scores of BI (five studies; SMD: 0.53; 95% CI: −0.09 to 1.16) (Figure 4A) and TLA (two studies; MD: −0.23; 95% CI −0.61 to 0.15) (Figure 4B). The I^2^ statistics were 88 and 0%, respectively, suggesting substantial heterogeneity between the studies reporting BI scores. However, this heterogeneity was resolved in the subgroup analysis of those studies in which the time since stroke onset was three days or less (SMD: 0.06; 95% CI: −0.29 to 0.40; I^2^ = 0%) (Appendix A). In contrast, subgroup analysis of those studies that initiated bloodletting within three days from onset showed a significant improvement in BI score, although heterogeneity across studies was still substantial (SMD: 0.85; 95% CI: 0.01 to 1.69; I^2^ = 89%) (Appendix A).

#### 3.4.3. Motor Function

For motor function, MA was conducted based on the scores of the three subdomains of the FMA. Bloodletting improved the motor function measured using FMA in the upper extremities (two studies; MD: 12.20; 95% CI: 9.67 to 14.73; I^2^ = 0%) (Figure 5A), lower extremities (three studies; MD: 3.86; 95% CI: 2.16 to 5.56; I^2^ = 40%) (Figure 5B), and hand (two studies; MD: 2.79; 95% CI: 0.06 to 5.53; I^2^ = 71%) (Figure 5C). However, the results of the subgroup analysis showed that while the effect of improving lower extremity motor function via bloodletting was maintained when the treatment was performed at least once a day (MD: 4.54; 95% CI: 3.26 to 5.82), this effect was no longer significant when the bloodletting was performed every other day (MD: 1.48; 95% CI: −1.90 to 4.86) (Appendix A).

#### 3.4.4. Safety

For the safety assessment, the pooled estimate of the data from studies reporting AEs is presented in Figure 6. There was no significant difference in the number of AEs reported between the bloodletting and non-bloodletting groups (two studies; RR: 0.91; 95% CI: 0.44 to 1.91; I^2^ = 0%).

#### 3.4.5. Publication Bias

Assessment of publication bias was possible only for the TER of neurological deficits, using data from more than 10 RCTs. The funnel plot did not show a visually obvious asymmetry (Figure 7), and Egger’s regression test did not indicate statistically significant evidence of publication bias, with a *p*-value of 0.3823.

#### 3.4.6. Sensitivity Analysis

The direction of the MA results did not change significantly, although some of the included studies were omitted (Appendix A). This indicates that the MA results were statistically robust.

### 3.5. Certainty of Evidence

The certainty-of-evidence evaluated results using the GRADE methodology are summarized in Table 2. The certainty of evidence for all outcomes was low or very low. The common causes of the overall downgrade were a high RoB and the small sample sizes of the included studies. In addition, when statistical heterogeneity among studies was significant (Higgins’ I^2^ ≥ 75%) or the 95% CIs of the MA results overlapped with the invalid interval, it was further downgraded. Consequently, the RoB domains for all outcomes were assessed as very serious, and the domains of inconsistency and imprecision were downgraded in many cases. Therefore, the overall certainty of the evidence derived from this study was mostly very low. Even when the supporting data size was large (CSS and TER of neurological deficits), the certainty of evidence was low.

## 4. Discussion

### 4.1. Significance of the Review and Comparison with Previous Studies

This study demonstrated that bloodletting performed on acupoints in the context of traditional East Asian medicine is effective and safe for the recovery of patients with acute stroke receiving conventional Western medicine treatments. A previous SR study conducted by Chen et al. [19] reported results similar to ours. Unlike a previous study [19], which included RCTs on the effect of bloodletting performed on specific acupoints (Jing-Well acupoints) published in journal articles or reported as dissertations up to 2015, we did not restrict the type of acupuncture and only included journal articles and more recently published articles. As a result, for the TER of neurological deficits, the only outcome commonly presented in the MA of our study and Chen’s [19], we derived a slightly larger effect size (fourteen studies; RR: 1.20; 95% CI: 1.14 to 1.26) than that of Chen’s study (seven studies; RR: 1.14; 95% CI: 1.07 to 1.22) [19].

Unlike the previous SR [19], which combined the changes in NIHSS and CSS scores before and after treatment (five studies; MD: 2.81; 95% CI: 1.71 to 3.91; I^2^ = 4%), we synthesized the post-treatment measurements and presented the NIHSS and CSS scores separately (NIHSS: three studies; MD: −2.08; 95% CI: −3.13 to −1.02; CSS: eight studies; MD: −4.15; 95% CI: −4.59 to −3.71) as another neurological deficit assessment measure. Given that a potentially clinically relevant change in the globally accepted NIHSS is considered to be two or more points [42], the findings of our study (MD −2.08) suggest that the neurological improvement effect of bloodletting is statistically and clinically significant. Even when we attempted to synthesize NIHSS and CSS data as in the previous SR (five studies; MD: 2.81; 95% CI: 1.71 to 3.91; I^2^ = 4%) [19], the benefit of bloodletting was still maintained (eleven studies; STD: −0.85; 95% CI: −1.18 to −0.53; I^2^ = 83%; MD: −3.63; 95% CI: −4.23 to −3.02; I^2^ = 33%).

We also presented the MA results for new outcomes that were not addressed in the previous SR [19]. The findings of our study suggested that bloodletting statistically significantly improved the motor function of the upper (MD: 12.20; 95% CI: 9.67 to 14.73) and lower extremities (MD: 4.15; 95% CI: 2.95 to 5.35) and hands (MD: 2.79; 95% CI: 0.06 to 5.53) as assessed using FMA subscores. In particular, the subscores of the upper and lower extremities were within the generally accepted range of minimally clinically important difference, which is 4.0–12.4 points [43]. This is inconsistent with the results of another RCT [12] that evaluated the efficacy of single-session bloodletting as an emergency treatment for patients with acute stroke and impaired consciousness. This previous study [12] reported that bloodletting may be most effective in improving the level of consciousness as measured according to the change in the total score in patients with moderate impairment. It [12] also showed that when evaluating the subscores of the Glasgow Coma Scale (GCS), eye opening and language responses were still effective, whereas motor responses did not change significantly. The results of this RCT [12] and our SR suggest that a single session of bloodletting may be insufficient to improve the motor response and that repeated procedures are necessary.

As another novel outcome, we also presented the results of an evidence synthesis on the effect of bloodletting on ADL performance. Unlike the outcomes related to neurological deficits or motor function, for which the clinical benefits of bloodletting were identified, ADL performance measured using BI (SMD: 0.53; 95% CI: −0.09 to 1.16) and TLA (MD: −0.23; 95% CI: −0.61 to 0.15) did not significantly differ with bloodletting.

Regarding safety, which was only described narratively in the previous SR [19], we were able to determine that bloodletting did not significantly affect the occurrence of AEs (RR: 0.91; 95% CI: 0.44 to 1.91). This finding suggests that bloodletting may be a clinically safe intervention, unlike previous reports that long-term or massive bleeding treatments in poorly controlled settings may cause complications such as infection [44] and anemia [45]. However, since the evidence supporting our SR was derived from only a limited number of small-sized studies, further studies with a larger number of cases are required to draw more robust conclusions about the safety of bloodletting.

Our SR also explored the potential factors that may have influenced the MA results on the effect of bloodletting through subgroup analysis. For example, the significant improvement in the NIHSS and CSS scores with bloodletting was maintained only in the subgroup that experienced a treatment duration of more than one week, whereas the effect was no longer significant when the treatment duration was one week or less. Furthermore, the subgroup analysis of the NIHSS scores suggested that the benefit of bloodletting was maintained only when the total number of sessions was more than ten, whereas it was no longer significant after ten or fewer sessions. The effects of bloodletting on lower extremity motor function appeared to be maintained only when the treatment interval was short. These findings suggest the following dosage criteria for optimizing the effect of bloodletting: once daily, at least five times a week, for more than one week, and more than ten total sessions.

Subgroup analysis according to stroke type showed that the significant benefit of bloodletting in terms of the TER of neurological deficits was maintained in patients with ischemic stroke. In contrast, in the subgroup that included patients with hemorrhagic stroke, the effect of bloodletting on improving TER was marginal, suggesting that bloodletting may be more beneficial for patients with ischemic stroke.

As mentioned above, in the main analysis, bloodletting did not have a significant effect on improving ADL function; however, the analysis of subgroups in which the duration of stroke was less than three days suggested that bloodletting may provide benefits to these patients. However, the aforementioned statements are not conclusive and require further research, as each subgroup included only a small number of studies, sometimes even one or two RCTs.

### 4.2. Underlying Mechanisms

Bloodletting, as defined in our SR, is similar to acupuncture in that it involves physical stimulation of the skin at acupoints by pricking it with a sharp needle tip. Acupuncture is a well-known intervention of traditional East Asian medicine that is considered a promising treatment option for acute stroke. Previous clinical studies have demonstrated that acupuncture provides clinical benefits, including recovery from neurological deficits, in patients in the acute and subacute stages [46].

However, bloodletting and acupuncture cannot be considered the same technique in that bloodletting does not involve the essential procedures of acupuncture, such as retaining the inserted needles for minutes and needle manipulation called *deqi.* Previous studies comparing the effects of bloodletting and acupuncture reported differential physiological responses between the two methods. For example, in a study on headaches, both interventions reduced pain, but only the bloodletting group had decreased nitric oxide and endothelin levels [47]. Research on acute brain injury models also demonstrated these differences [17]. While acupuncture stabilized blood pressure and increased intracranial blood perfusion, bloodletting was reported to improve focal brain tissue hemorrhage and cerebral edema by promoting coagulation factor synthesis [17].

In traditional East Asian medicine, bloodletting is known to regulate the circulation of *qi* and blood and unblock meridians—the channels of *qi* and blood [48]. Recent studies have revealed that bloodletting improves hemodynamics, suppresses inflammation, and modulates coagulation factors [49]. These mechanisms are also applicable to acute brain injury situations: bloodletting contributes to neurological recovery by repairing blood–brain barrier damage and activating glial cells [17,50].

### 4.3. Limitations of This Review

Only a few studies with small sample sizes were included in the analyses. In general, conclusions drawn from smaller datasets are likely to have lower statistical power and higher sampling error [51]. Recently published studies have gradually increased in size, involving more than 100 participants. However, none of the included studies provided information as to whether they applied the optimal sample size calculated in advance during the study design stage.

There is also a significant risk of regional and language biases. This is because only studies conducted in mainland China and published in Chinese in domestic journals were included.

The RoB 2 evaluation results suggested that the reporting quality of all studies included in our SR was generally low. In particular, most or all of the included studies omitted descriptions of the randomization and allocation concealment procedures and did not provide preplanned study protocols. Although blinding of practitioners and patients may be difficult due to the inherent nature of bloodletting, even information on the blinding of assessors, which is essential for the reliability of outcome assessments, was not provided. As a result, the overall RoB of all included studies was high.

For the above reasons, the certainty of the evidence in our SR assessed using the GRADE approach was very low or low. This means that it is difficult to be certain about the consistency between the true effect of bloodletting and the estimates derived from our SR. To draw more robust conclusions based on higher certainty of evidence, larger-scale trials conducted in more diverse locations with a lower risk of bias and better reporting quality using more rigorous methodologies are warranted.

We only performed an MA on neurological deficits, ADL function, motor function, and overall safety in patients with acute stroke; therefore, this SR does not provide any information on other outcomes. None of the included studies provided data on mortality or stroke recurrence among the outcomes we intended to include as secondary endpoints. Additionally, this SR does not explain the effects of frequent complications in patients with stroke, such as dysphagia, dysarthria, aphasia, cognitive impairment, and dysesthesia. In addition, we only addressed the phenomenon observable in clinical settings, and further research is required on the underlying mechanism of bloodletting in acute stroke. This SR included only RCTs that evaluated the post-treatment effects of repeated bloodletting therapy performed over two or more sessions. Therefore, to determine the immediate effect of a single session of bloodletting, we need to refer to other studies [12] in addition to our SR.

Further studies are needed to determine whether bloodletting is still effective in patients in the recovery or chronic phases after the acute phase. In addition, the long-term effects of bloodletting require further evaluation because this SR only examined post-treatment measurements, which lasted five weeks at most. Finally, although we attempted to conduct a comprehensive and systematic literature search to achieve the objectives of our study, we cannot rule out the possibility that there may be evidence that was not detected using the search strategy of our study.

## 5. Conclusions

These findings indicate that bloodletting may be an effective and safe therapeutic option for the recovery of patients with acute stroke. Currently available evidence indicates that the addition of bloodletting in patients receiving conventional treatment with Western medicine for acute stroke significantly improves neurological deficits and motor function and does not significantly increase the frequency of AEs. Bloodletting does not improve ADL performance; however, in a limited group of patients with a stroke onset of less than three days, bloodletting was effective. The subgroup analysis results provide more clues to several factors that may influence the effectiveness of bloodletting: types of stroke (ischemic stroke) associated with improvement in the TER in neurological deficits, sufficiently high intervention doses (treatment period > one week and total sessions > ten times) for improvement in neurological deficits measured using the NIHSS or CSS, and sufficiently frequent intervention (number of treatment session ≥ five days per week) for enhancing the motor power of lower extremities.

However, given that this SR was based on a small number of studies with a high RoB and that the overall quality of evidence was low, readers should be cautious when attempting to extrapolate the results to the general population. To confirm the findings of this study, larger-sized, higher-quality studies conducted in more diverse locations are warranted.

## Figures and Tables

**Figure 1 healthcare-12-02060-f001:**
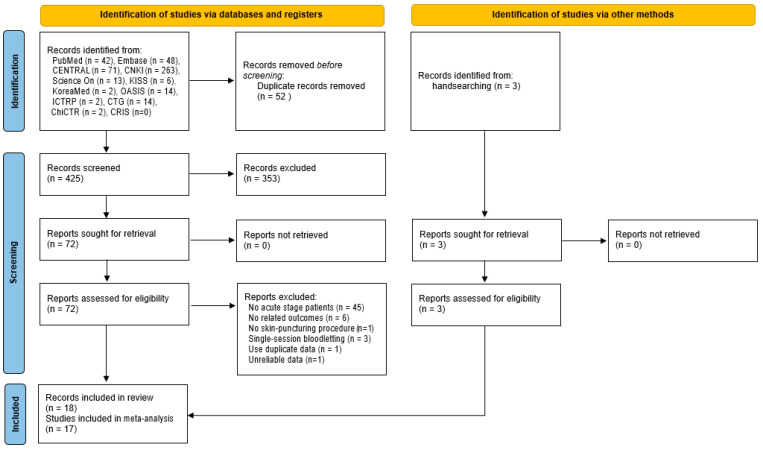
Flow diagram of the study selection.

**Figure 2 healthcare-12-02060-f002:**
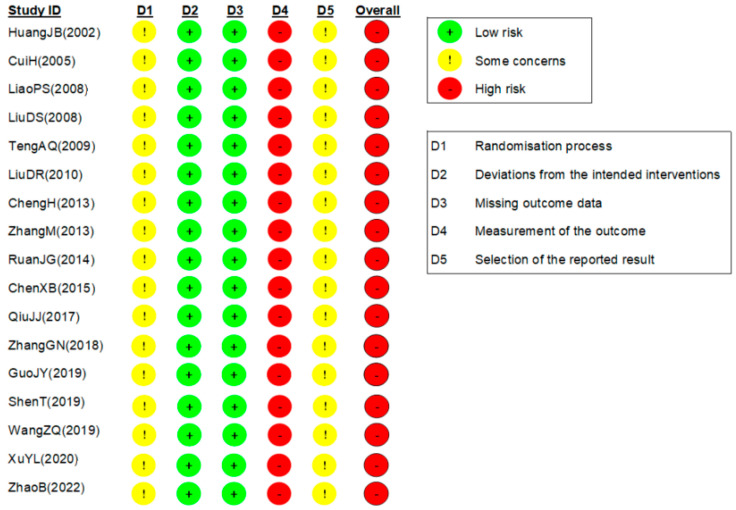
Risk of bias of the included studies [25,26,27,28,29,30,31,32,33,34,35,36,37,38,39,40,41].

**Figure 3 healthcare-12-02060-f003:**
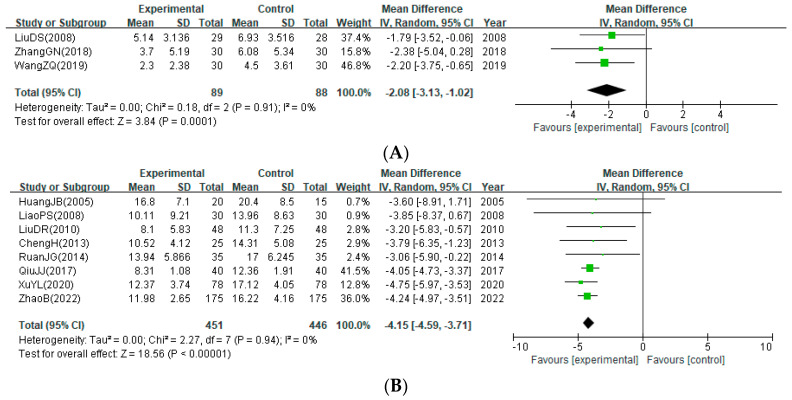
Improvement in neurological deficits. (**A**) Post-treatment score of the National Institute of Health Stroke Scale; (**B**) post-treatment score of the Chinese Stroke Scale; (**C**) treatment effective rate for neurological deficits [25,26,27,28,29,30,31,32,33,35,36,38,39,40,41].

**Figure 4 healthcare-12-02060-f004:**
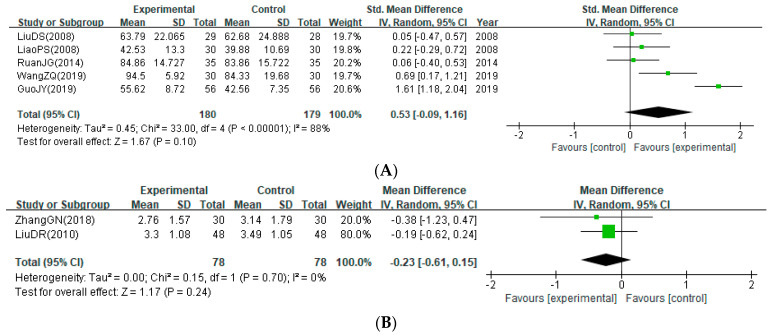
Improvement of activities of daily living function. (**A**) Post-treatment score of Barthel index; (**B**) post-treatment Total Life Ability score [25,27,28,30,33,36,37].

**Figure 5 healthcare-12-02060-f005:**
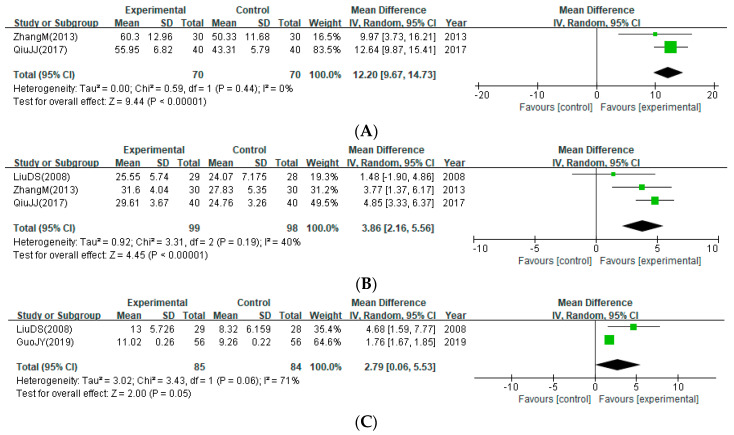
Improvement of motor function assessed using the post-treatment score of the Fugl-Meyer Assessment Scale (FMA). (**A**) FMA (upper extremities); (**B**) FMA (lower extremities); (**C**) FMA (hands) [28,32,35,37].

**Figure 6 healthcare-12-02060-f006:**
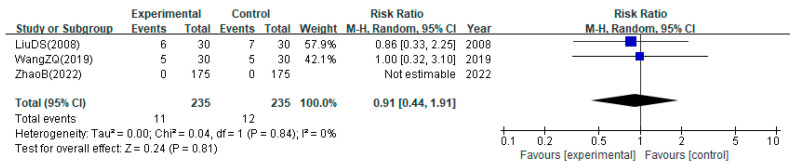
Safety assessed using the frequency of adverse events [28,35,41].

**Figure 7 healthcare-12-02060-f007:**
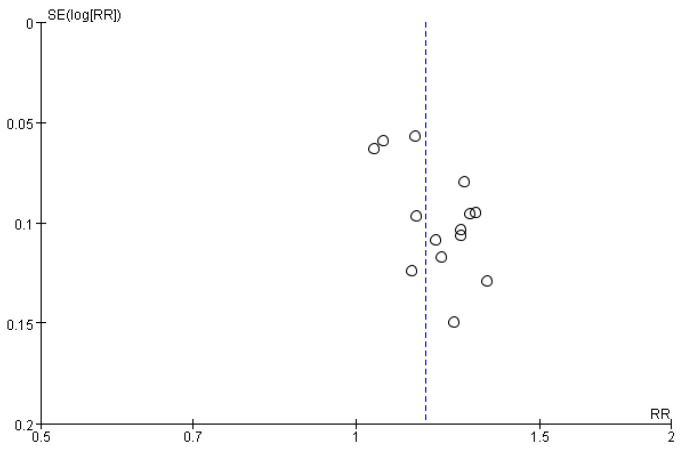
Funnel plot.

**Table 1 healthcare-12-02060-t001:** Characteristics of the included studies.

Author (yr)	Participants				Intervention		Details of BL		
	No. (M/F)	Age ^1^	Stroke Type	TSO	TG	CG	Tx Acupoints	Tx Tool	Bleeding Amount
Huang JB (2005) [26]	35 (18/17)	54–74	IS	≤48 h	BL + WM	WM	Total 4–6, 1–2 points per group (JW(Hd), Ex-UE 11, ear tip and dorsal auricular veins)	TEAN	2–3 drops
Cui H (2005) [39]	90 ^2^ (51/39)	TG: 53, CG: 54	IS	≤5 d	BL + Acu	Acu	JW (Hd or Ft)	TEAN	5–7 drops
Liao PS (2008) [27]	60 ^3^ (31/29)	TG: 67, CG: 68	IS	24 h–3 d	BL + Acu + WM	Acu+WM	JW (Hd)	DAN	2–3 drops
Liu DS (2008) [28]	60 (32/28)	TG: 61, CG: 65	IS	≤10 d	BL + Acu	Acu	JW (HdFt)	TEAN or DAN	1–3 drops
Teng AQ (2009) [29]	50 (38/12)	TG: 56, CG: 54	IS	NR	BL + WM	WM	JW (HdFg)	TEAN	NR
Liu DR (2010) [30]	96 (57/39)	44–76	IS	1–21 d	BL + HM + WM	HM+WM	JW (Hd)	TEAN	1–3 drops
Cheng H (2013) [31]	50 (22/28)	42–80	IS, HS	≤72 h	BL + WM	WM	JW (Hd)	DAN	1 drop
Zhang M (2013) [32]	60 ^4^ (37/23)	18–77	TIS	≤24 h	BL + HM + WM	HM+WM	JW (Hd)	TEAN	3 drops
Ruan JG (2014) [33]	70(40/30)	TG: 52, CG: 52	IS	≤7 d	BL + Acu + WM	Acu+WM	6–8 acupoints among JW(HdFt)	TEAN	10 drops
Chen XB (2015) [34]	60 (43/17)	TG: 56, CG: 54	IS	≤6 h	BL + WM	WM	PC9 of JW	TEAN	2–3 drops
Qiu JJ (2017) [35]	80 (43/37)	58–69	IS	≤4 d	BL + WM	WM	Ear tip	DAN	≤ 5 ml
Zhang GN (2018) [36]	60 ^4^ (32/28)	42–73	IS	≤1 d	BL + HM	HM	12 JW, GV20, EX-HN1	TEAN	3 drops
Guo JY (2019) [37]	112 (67/45)	60–79	IS	≤44 h	BL + WM	WM	Ex-UE 11	TEAN	5–8 drops
Shen T (2019) [38]	80 ^5^ (50/30)	TG: 68, CG: 69	HS	>72 h ^6^	BL + WM	WM	JW	TEAN	3 drops
Wang ZQ (2019) [25]	60 (37/23)	48–79	IS	<24 h	BL + WM	WM	KI1 and HT9 of JW, Ex-UE 11, GV20	TEAN	3–5 drops
Xu YL (2020) [40]	156 (87/69)	47–78	IS	6–72 h	BL + HM	HM	JW (HdFt)	TEAN	NR
Zhao B (2022) [41]	350 (204/146)	45–85	IS	8–72 h	BL + HM	HM	JW (Hd)	DAN	NR
**Author (yr)**	**Details of BL**				**Outcome Measures**				
	**Tx Interval ^7^**	**Total Session**	**Period (d)**		**Neurological** **Deficit**	**Functional** **Independency**	**Motor Function**		**Safety**
Huang JB (2005) [26]	None	7	7		CSS				
Cui H (2005) [39]	None	14	14		TER				
Liao PS (2008) [27]	2 d/wk	10	14		CSS	BI			
Liu DS (2008) [28]	4 d/wk	12	28		NIHSS, TER	BI	FMA (m, L, A, W, H)		AE
Teng AQ (2009) [29]	E.O.D. from 6th d	10	15		TER				
Liu DR (2010) [30]	1 d/11 d	20-	21		CSS, TER	TLA			
Cheng H (2013) [31]	None	14	14		CSS, TER				
Zhang M (2013) [32]	None	84	28		TER		FMA(U,L)		
Ruan JG (2014) [33]	2 d/wk	15	21		CSS, TER	BI			
Chen XB (2015) [34]	NR	NR (≤10)	10		TER				
Qiu JJ (2017) [35]	1 d/wk	12	14		CSS, TER		FMA(U,L), MP(U.L)		
Zhang GN (2018) [36]	2 d/wk	5	7		NIHSS, TER	TLA			
Guo JY (2019) [37]	1 d/wk	24	28		CSS(H)	MBI	FMA(H), CSS(H)		
Shen T (2019) [38]	2 d/3 d	12	36		NIHSS(D), TER	TLA(D)	FMA(D)		
Wang ZQ (2019) [25]	None	14	14		NIHSS, TER	BI, FCA			AE
Xu YL (2020) [40]	E.O.D. from 6th d	10	15		CSS, TER	TLA (description only)			
Zhao B (2022) [41]	E.O.D. from 6th d	9	14		CSS, TER				AE

Notes. ^1^: For age column, the age distribution of participants was preferentially described, but if it was not available, the average value was described. ^2^: Three-armed study (A: BL at JW (H) + Acu vs. B: BL at JW + Acu vs. C: Acu), but A and B were combined and treated as one Tx group in our study (A + B vs. C). ^3^: Three-armed study (A: BL + Acu + WM vs. B: Acu + WM vs. C: WM), but only data from two groups are included in our study (A vs. B). ^4^: Three-armed study (A: BL + HM + WM vs. B: HM + WM vs. C: BL + WM), but only data from two groups are included in our study (A vs. B). ^5^: Three-armed study (A: BL + WM + hyperbaric oxygen vs. B: BL + WM vs. C: WM), but only data from two groups are included in our study (B vs. C). ^6^: BL was performed on patients who underwent hematoma removal within 24 h of the onset of cerebral hemorrhage and whose vital signs were stable for 48 h after surgery. Therefore, it is assumed that BL was performed on patients at least 72 h after onset. ^7^: Tx interval indicates period of treatment break. Abbreviations. A: arm, Acu: acupuncture, AE: adverse event, BI: Barthel Index, BL: bloodletting, CG: control group, CSS: Chinese Stroke Scale, d: day, D: delta, DAN: disposable acupuncture needle, E.O.D: once a day and then every other day, Ex-HN1: *Sishencong* acupoints (四神聰), Ex-UE 11: *Shixuan* acupoints (十宣穴), F: female, FCA: Functional Comprehensive Assessment, FMA: Fugl-Meyer Assessment, Ft: foot, GV20: *Baihui* acupoints (百會穴), h: hours, Hd: hand, HT9: *Shaochong* (少沖穴), HM: herbal medicine, HS: hemorrhagic stroke, IS: ischemic stroke, JW: *Jing*-Well acupoints (手井穴), KI1: *Yongquan* (涌泉穴), L: lower extremities, M: male, m: motor, MBI: modified Barthel Index, NIHSS: National Institute of Health Stroke Scale, No.: number, NR: not reported, PC9: *Zhongchong* acupoints (中衝穴), TEAN: three-edged acupuncture needle, TER: total effective rate, TG: treatment group, TIS: traumatic ischemic stroke, TLA: Total Life Ability score, TSO: time since onset, Tx: treatment, U: upper extremities, W: wrist, wk: week, WM: Western medicine, Yr: year.

**Table 2 healthcare-12-02060-t002:** Certainty of evidence.

Outcome Measures		No. P (S)	RoB	Inconsistency	Indirectness	Imprecision	Publication Bias	Overall Certainty of Evidence	Anticipated Absolute Effects
NE	NIHSS	177 (3)	Very serious ^1^	Not serious	Not serious	Serious ^3^	NA	⊕◯◯◯ Very low	MD 2.08 lower (3.13 lower to 1.02 lower)
	CSS	897 (8)	Very serious ^1^	Not serious	Not serious	Not serious	NA	⊕⊕◯◯ Low	MD 4.15 lower (4.59 lower to 3.71 lower)
	TER	1319 (14)	Very serious ^1^	Not serious	Not serious	Not serious	None	⊕⊕◯◯ Low	151 more per 1000 (from 106 more to 196 more)
FI	BI	359 (5)	Very serious ^1^	Very serious ^2^	Not serious	Very serious ^3,4^	NA	⊕◯◯◯ Very low	SMD 0.53 higher (0.09 lower to 1.16 higher)
	TLA	156 (2)	Very serious ^1^	Very serious ^2^	Not serious	Very serious ^3,4^	NA	⊕◯◯◯ Very low	MD 0.23 lower (0.61 lower to 0.15 higher)
MF	FMA (UE)	140 (2)	Very serious ^1^	Not serious	Not serious	Serious ^3^	NA	⊕◯◯◯ Very low	MD 12.2 higher (9.67 higher to 14.73 higher)
	FMA (LE)	197 (3)	Very serious ^1^	Not serious	Not serious	Serious ^3^	NA	⊕◯◯◯ Very low	MD 4.15 higher (2.95 higher to 5.35 higher)
	FMA (H)	169 (2)	Very serious ^1^	Serious ^2^	Not serious	Serious ^3^	NA	⊕◯◯◯ Very low	MD 2.79 higher (0.06 higher to 5.53 higher)
Safety	AE	120 (2)	Very serious ^1^	Not serious	Not serious	Very serious ^3,4^	NA	⊕◯◯◯ Very low	18 fewer per 1000 (from 112 fewer to 182 more)

^1^: The overall risk of bias of the included studies was high. ^2^: I^2^ ≥ 75%. ^3^: Number of participants < 400. ^4^: The 95% confidence interval of meta-analysis results overlapped with the invalid interval. Abbreviations. AE: adverse event, BI: Barthel Index, CSS: Chinese Stroke Scale, FMA: Fugl-Meyer Assessment, H: hand, LE: lower extremities, MD: mean difference, NIHSS: National Institute of Health Stroke Scale, SMD: standardized mean difference, TER: total effective rate, TLA: Total Life Ability scale.

## Data Availability

The data supporting the findings of this study are available from the corresponding author upon request.

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
