# Peer review of "Bloodletting for Acute Stroke Recovery: A Systematic Review and Meta-Analysis"

_healthcare, 2024, doi:10.3390/healthcare12202060_

Round 1
Reviewer 1 Report
Comments and Suggestions for Authors
1. Introduction, page 1, line 52: How much blood is removed during a typical bloodletting session and how often is it repeated?
2. Methods, page 2, line 85: What does the loss of a few drops of blood add to traditional acupuncture?
3. Methods, page 2, line 92: Did patients undergo thrombolytic therapy if eligible?
4. Results, page 5, line 202: What was included under standard Western medicines?
5. Results, page 5, line 216: When is bloodletting started after stroke symptoms start?
6. Discussion: What is the proposed mechanism of action of bloodletting? Since most Western readers will not be familiar with this practice, it would be helpful to describe it in greater detail.
Comments on the Quality of English Languagemoderate revisions needed
Author Response
|
Response to Reviewer 1 Comments
|
||
|
1. Summary |
|
|
|
Thank you very much for taking the time to review this manuscript. Please find the detailed responses below and the corresponding revisions/corrections highlighted/in track changes in the re-submitted files.
|
||
|
2. Questions for General Evaluation |
Reviewer’s Evaluation |
Response and Revisions |
|
Does the introduction provide sufficient background and include all relevant references? |
Yes/Can be improved/Must be improved/Not applicable |
Please check the point-by-point responses below.
|
|
Is the research design appropriate? |
Yes/Can be improved/Must be improved/Not applicable |
|
|
Are the methods adequately described? |
Yes/Can be improved/Must be improved/Not applicable |
|
|
Are the results clearly presented? |
Yes/Can be improved/Must be improved/Not applicable |
|
|
Are the conclusions supported by the results? |
Yes/Can be improved/Must be improved/Not applicable |
|
|
3. Point-by-point response to Comments and Suggestions for Authors |
||
|
Comments 1: Introduction, page 1, line 52: How much blood is removed during a typical bloodletting session and how often is it repeated? |
||
|
Response 1: This is an important question regarding the specific method of bloodletting. Thank you for pointing this out. In the context of East Asian traditional medicine, bloodletting at acupoints is a method of releasing only few drops of blood by shallowly puncturing the skin at a certain point, unlike other similar treatments such as phlebotomy or venesection, which remove large amount of blood by cutting a blood vessel. However, as with many other traditional medical interventions, the amount of blood removed at one session or the frequency of the intervention has not been officially promised. Therefore, we thought it would be meaningful to inform readers of the actual applied bleeding amount and intervention frequency in the included studies, and thus extracted the relevant data and described them in the manuscript (Table 1 and page 6, lines 227-234). 1) Bleeding amount: As written in the ‘Bleeding amount’ column of Table 1 (page 7), at most 10 drops, and usually 3 drops of blood per an acupoint were removed. 2) Frequency: As written in the ‘Total session’ and ‘Period (d)’ columns of Table 1 (continued) (page 8), most cases were performed once a day for about 2 weeks. |
||
|
|
||
|
Comments 2: What does the loss of a few drops of blood add to traditional acupuncture? Response 2: Thank you for pointing this out. Bloodletting and acupuncture both use needles to stimulate acupoints but differ in technique: Bloodletting doesn’t involve needle retention or manipulation. The physiological responses induced by two treatments are also known to be distinct: While acupuncture improves cerebral blood perfusion and stabilizes blood pressure, bloodletting helps reduce cerebral edema and promotes coagulation. A new subsection (4.2. Underlying mechanisms) was added to the discussion section explaining the differences between these two treatments and the findings of previous studies on the mechanism of action of bloodletting (page 15, line 465-481).
Comments 3: Methods, page 2, line 92: Did patients undergo thrombolytic therapy if eligible? Response 3: Yes. For ethical reasons, all patients enrolled in the clinical trials received standard Western care for acute stroke, including IV thrombolysis if eligible, regardless of assigned group. We added sentences clarifying this (page 2, line 88 - page 3, line 92).
Comments 4: Results, page 5, line 202: What was included under standard Western medicines? Response 4: Thank you for pointing this out. This refers to the standard care generally prescribed in clinical practice guidelines for acute stroke in Western medicine, including general supportive care, medications to prevent and manage complications, nursing interventions, rehabilitation, IV thrombolysis, endovascular therapies, and surgical treatment. Sentences clarifying this have been added (page 2, line 88 - page 3, line 92).
Comments 5: Results, page 5, line 216: When is bloodletting started after stroke symptoms start? Response 5: Thank you for pointing this out. A sentence that clearly state that interventions began immediately after patients were enrolled in the most of the included studies (page 5, line 221-223). Most cases enrolled patients within 3 days of stroke onset, so it can be said that most of them began bloodletting within 3 days of stroke onset.
Comments 6: Discussion: What is the proposed mechanism of action of bloodletting? Since most Western readers will not be familiar with this practice, it would be helpful to describe it in greater detail. Response 6: We agree with this comment. Therefore, we added a subsection (4.2. Underlying mechanisms) introducing tentative mechanisms explored through preclinical studies, including brain edema reduction, coagulation regulation, and neuroprotection (page 15, lines 482-487).
|
||
|
4. Response to Comments on the Quality of English Language |
||
|
Point 1: Moderate revisions needed |
||
|
Response 1: The first draft was proofread by a professional English editing service provider, but the revised manuscript was proofread again by MDPI’s editing service. |
||
Reviewer 2 Report
Comments and Suggestions for Authors
Dear Authors,
thank you for your submission of the article "Bloodletting for Acute Stroke Recovery: A Systematic Review and Meta-Analysis." Your systematic review is thorough, and you have provided a well-rounded analysis of the effectiveness of bloodletting in acute stroke recovery. However, there are several areas that require clarification and enhancement to improve the overall quality of the manuscript.
1. Introduction: Clarify the novelty of your work compared to previous systematic reviews. While you mention gaps in previous studies, these should be better defined, particularly with respect to new outcome measures like NIHSS.
2. Methods: Your search strategy is generally comprehensive, but there is a lack of detail concerning the reasons why, certain forms of bloodletting were excluded.
3. Results: You present the results clearly, especially in terms of neurological deficits and motor function. However, the discussion around adverse events is underdeveloped, given the limited data available. More explanation is needed on how this impacts the conclusions drawn about the safety of bloodletting.
4. Discussion: The discussion would benefit from a more thorough exploration of the possible mechanisms behind the effectiveness of bloodletting, particularly in early stroke recovery. Additionally, the limitations section could be more specific, focusing on the high RoB and the lack of safety data.
5. Conclusion: Please ensure that the conclusion reflects the overall low certainty of evidence and the need for larger, high-quality studies to confirm your findings.
Best regards
Author Response
|
Response to Reviewer 2 Comments
|
||
|
1. Summary |
|
|
|
Thank you very much for taking the time to review this manuscript. Please find the detailed responses below and the corresponding revisions/corrections highlighted/in track changes in the re-submitted files.
|
||
|
2. Questions for General Evaluation |
Reviewer’s Evaluation |
Response and Revisions |
|
Does the introduction provide sufficient background and include all relevant references? |
Yes/Can be improved/Must be improved/Not applicable |
Please check the point-by-point responses below.
|
|
Is the research design appropriate? |
Yes/Can be improved/Must be improved/Not applicable |
|
|
Are the methods adequately described? |
Yes/Can be improved/Must be improved/Not applicable |
|
|
Are the results clearly presented? |
Yes/Can be improved/Must be improved/Not applicable |
|
|
Are the conclusions supported by the results? |
Yes/Can be improved/Must be improved/Not applicable |
|
|
3. Point-by-point response to Comments and Suggestions for Authors Thank you for your submission of the article "Bloodletting for Acute Stroke Recovery: A Systematic Review and Meta-Analysis." Your systematic review is thorough, and you have provided a well-rounded analysis of the effectiveness of bloodletting in acute stroke recovery. However, there are several areas that require clarification and enhancement to improve the overall quality of the manuscript.
|
||
|
Comments 1: Introduction: Clarify the novelty of your work compared to previous systematic reviews. While you mention gaps in previous studies, these should be better defined, particularly with respect to new outcome measures like NIHSS. |
||
|
Response 1: Thank you for pointing this out. Differences between the previous SR and this study are described in detail in a subsection of Discussion (4.1. Significance of the review and comparison with previous studies, page 13 - 14). However, in response to your advice, we added sentences to the last two paragraphs of the Introduction to further elaborate on the novelty of this study compared to previous SR, such as the inclusion of more recently published RCTs, the absence of restrictions on the type of acupoints used, and the application of more universally accepted outcomes (page 2, line 64 -76). |
||
|
Comments 2: Your search strategy is generally comprehensive, but there is a lack of detail concerning the reasons why, certain forms of bloodletting were excluded. |
||
|
Response 2: We agree with this comment. We excluded other modalities that also induce blood release, such as venesection, phlebotomy, hemodilution, Hijama, or wet cupping, because they differ significantly from bloodletting as defined in our study in terms of the interventional procedures: Cutting a blood vessel to remove substantial amount of blood vs shallowly puncturing the skin with a small needle to release few drops of blood; Targeting a certain area and adding negative pressure by cupping in addition to puncturing vs targeting a certain point and simply puncturing the skin. Sentences explaining this difference in more detail were added (page 3, line 96 – 105).
Comments 3: You present the results clearly, especially in terms of neurological deficits and motor function. However, the discussion around adverse events is underdeveloped, given the limited data available. More explanation is needed on how this impacts the conclusions drawn about the safety of bloodletting. Response 3: We agree with this comment. So we added a separate paragraph interpreting the meta-analysis results on the safety of bloodletting explaining that the result of our study suggest the safety of bloodletting, but the supporting evidence is limited, so we cannot draw definitive conclusions (page 4, line 433-440).
Comments 4: Discussion: The discussion would benefit from a more thorough exploration of the possible mechanisms behind the effectiveness of bloodletting, particularly in early stroke recovery. Additionally, the limitations section could be more specific, focusing on the high RoB and the lack of safety data. Response 4: Thank you for pointing these important issued out. 1) Possible mechanisms: The mechanism of action of bloodletting has not been clearly elucidated. However, a subsection titled 4.2. Underlying mechanisms (page 15) was added to the Review, which supplemented the description of the similarities and differences between acupuncture and bloodletting, and the findings of previous preclinical studies that explored the mechanisms involved in the effect of bloodletting on acute brain injury. 2) High RoB: The paragraphs in the limitations subsection were separated to improve readability (page 15-16). The assessment results of each domain of RoB 2 tool are described in the 3rd paragraph of the limitations section (page 15, line 499-505). A sentence describing the high overall RoB of the included studies was added to the end of the paragraph (page 14, line 505). 3) Lack of safety data: We added a separate paragraph interpreting the meta-analysis results on safety, and included a sentence within this paragraph describing the lack of data supporting the safety of bloodletting (page 14, line 438-439).
Comments 5: Please ensure that the conclusion reflects the overall low certainty of evidence and the need for larger, high-quality studies to confirm your findings. Response 5: Thank you for pointing this out. To improve readability, we separated the paragraphs describing the findings of this study and the further challenges to improve the current limitations. We also revised the sentences using the expressions you recommended that the findings of this study are based on a small number of small-sized studies with a high risk of bias and, as a result, have produced evidence of generally low quality (page 16, line 545-549).
|
||
|
4. Response to Comments on the Quality of English Language |
||
|
Point 1: None |
||
|
Response 1: The first draft was proofread by a professional English editing service provider, but the revised manuscript was proofread again by MDPI’s editing service. |
||
Round 2
Reviewer 1 Report
Comments and Suggestions for Authors
I believe appropriate modifications have been made
Comments on the Quality of English Languageadequate